# Effects of Chlorogenic Acid on Performance, Anticoccidial Indicators, Immunity, Antioxidant Status, and Intestinal Barrier Function in Coccidia-Infected Broilers

**DOI:** 10.3390/ani12080963

**Published:** 2022-04-08

**Authors:** Huawei Liu, Peng Chen, Xiaoguo Lv, Yingjun Zhou, Xuemin Li, Shengnan Ma, Jinshan Zhao

**Affiliations:** 1College of Animal Science and Technology, Qingdao Agricultural University, Qingdao 266109, China; liuhuawei@qau.edu.cn (H.L.); chensky9995@163.com (P.C.); lvxiaoguo1995@163.com (X.L.); lxm5968@126.com (X.L.); 17852023061@163.com (S.M.); 2Qingdao Vland Biotech Group Co., Ltd., Qingdao 266000, China; zhouyj@vlandgroup.com

**Keywords:** antioxidant status, broiler, chlorogenic acid, coccidiosis, immunity, intestinal barrier function

## Abstract

**Simple Summary:**

Coccidiosis impairs the growth performance, intestinal health, and immunity of broilers, resulting in significant economic losses to the poultry industry. The use of anticoccidial drugs can lead to the emergence of resistant strains and drug resistance, and the control effect of vaccines is limited. Thus, there is an urgent need to develop substitutes for anticoccidial drugs. Chlorogenic acid (CGA) was reported to have positive effects on the performance and intestinal health of broilers. However, little is known about the application and mechanism of CGA in coccidia-infected broilers. The results of this study showed that dietary supplementation with CGA exhibited anticoccidial activities, and improved growth performance, immunity, antioxidant status, and intestinal barrier function in coccidia-infected broilers.

**Abstract:**

The effects of chlorogenic acid (CGA) on growth performance, anticoccidial indicators (oocysts per gram of excreta, cecal lesion score, and bloody diarrhea score), immunity, antioxidant status, and intestinal barrier function in coccidia-infected broilers were investigated. A total of 240 one-day-old Arbor Acres broilers were randomly divided into four groups with six replicates of ten broilers each for 42 days. Four treatments included control diet (non-infected control, NC), control diet +*Eimeria* infection (infected control, IC), control diet +0.5 g/kg CGA + *Eimeria* infection (CGA0.5), and control diet +1 g/kg CGA + *Eimeria* infection (CGA1). At day 14, each broiler in IC, CGA0.5, and CGA1 groups was orally inoculated with 1 mL saline containing 4 × 10^5^ sporulated oocysts. The results showed that the CGA1 group increased the average daily gain by 12.57% (*p* < 0.001) and decreased the feed/gain ratio (*p* = 0.010) and mortality (*p* = 0.030) by 13.00% and 77.76%, respectively, of broilers from 14 to 42 days compared with the IC group. The CGA1 group decreased the oocysts per gram of excreta (*p* < 0.001) and bloody diarrhea score (*p* = 0.001) compared with the IC group. The CGA0.5 and CGA1 groups increased total antioxidant capacity (*p* < 0.001) at day 21 and villus height (*p* < 0.001) in the duodenum and jejunum at day 42, and decreased the levels of interleukin 6 (IL-6) (*p* = 0.002), malondialdehyde (MDA) (*p* < 0.001), *D*-lactic acid (*p* < 0.001), and diamine oxidase (DAO) (*p* < 0.001) at day 21 and the levels of MDA (*p* < 0.001) and *D*-lactic acid (*p* = 0.003) at day 42 compared with the IC group. In the CGA1 group, villus height in the duodenum (*p* < 0.001) and jejunum (*p* = 0.017) increased at day 21 and in the ileum (*p* < 0.001) at day 42, and the level of DAO (*p* < 0.001) decreased at day 42 compared with the IC group. Broilers in the IC group had a higher IL-6 level (*p* = 0.048) at day 42 and lower IL-10 (*p* = 0.027) and immunoglobulin A (*p* = 0.042) levels at day 21, and IL-10 level (*p* = 0.017) at day 42 than those in the NC group, while no significant differences were observed among the NC, CGA0.5, and CGA1 groups. In conclusion, dietary supplementation with 1 g/kg CGA improved growth performance, immunity, antioxidant status, and intestinal barrier function in coccidia-infected broilers.

## 1. Introduction

The main coccidiosis species are *E. tenella*, *E. necatrix*, *E. acervulina*, and *E. maxima* [1], which can cause significant intestinal tissue damage, high mortality, and economic losses in the poultry industry [2,3]. Current prevention and treatment of coccidiosis still involve drugs and vaccines [4]. However, the use of anticoccidial drugs can lead to the emergence of resistant strains and drug resistance, and the control effect of vaccines is limited [5,6,7,8,9]. Thus, there is an urgent need to develop substitutes for anticoccidial drugs.

Chlorogenic acid (CGA), an ester of caffeic acid and quinic acid, is a major phenolic compound in coffee, which has high safety and mature extraction technology [10]. It is widely found in Chinese herbs such as *Flos lonicerae* and *Eucommia ulmoides* [11]. Studies have shown that CGA has biological effects, such as antibacterial [12], antioxidant [13,14], and anti-inflammatory [15]. Thus, CGA is recommended as a nutritional supplement for animals [16]. Dietary supplementation with CGA can decrease the expression of pro-inflammatory factors (IFN-β, IFN-γ, IL-1, IL-17A, IL-22, and TNF-α) and improve the growth performance and intestinal injury in broilers with necrotic enteritis [17]. Our previous study demonstrated that dietary supplementation with CGA could improve the growth performance and increase the activity of antioxidant enzymes in heat-stressed broilers [18] and oxidatively-stressed broilers [19]. However, little is known about the application and mechanism of CGA in coccidia-infected broilers. This study was conducted to investigate the effects of dietary supplementation with CGA on growth performance, anticoccidial indicators (oocysts per gram of excreta, cecal lesion score, and bloody diarrhea score), immunity, antioxidant status, and intestinal barrier function in coccidia-infected broilers.

## 2. Materials and Methods

### 2.1. Ethical Approval

All procedures were approved by the Animal Care and Use Committee of Qingdao Agricultural University (Qingdao, China).

### 2.2. Materials

*Eimeria* (containing four *Eimeria* species of *E. tenella*, *E. necatrix*, *E. acervulina*, and *E. maxima*) used in the present study was isolated and provided by the Parasitology Laboratory, College of Veterinary Medicine, Qingdao Agricultural University, China. Oocysts were preserved in 2.5% potassium dichromate solution, and cultured at 28 °C for 48 h. The degree of oocysts sporulation was observed under microscope. When the proportion of spore oocysts exceeds 80%, store them in a refrigerator at 4 °C for later use [20]. The CGA (with a purity of 98%) was purchased from Changsha Biotechnology Co., Ltd. (Changsha, China). The experimental diet was provided by Henan Feed Company (Zhengzhou, China). The one-day-old Arbor Acres broilers were provided by Henan Animal Husbandry Company (Zhengzhou, China).

### 2.3. Animals and Experimental Design

A total of 240 one-day-old Arbor Acres broilers (initial body weight 33.94 ± 0.25 g) were weighed and randomly divided into four groups with six replicated cages of ten broilers each with a 42-day feeding period. There were no significant differences in initial body weight between the four treatment groups. The dimension of cages used in the study is 100 cm × 100 cm × 60 cm (length × width × high). Water and feed were provided ad libitum, with the photoperiod set at 23 L:1 D throughout the study. The groups were fed a control diet (non-infected control, NC), control diet + *Eimeria* infection (infected control, IC), control diet +0.5 g/kg CGA + *Eimeria* infection (CGA0.5), and control diet +1 g/kg CGA + *Eimeria* infection (CGA1). The basal diet was formulated to meet the requirements suggested by the National Research Council (NRC, 1994; Table 1). No coccidiostats or antibiotics were included in the diets. On day 14, all groups except the NC group were inoculated with 1 mL saline containing 4 × 10^5^ sporulated oocysts (*E. tenella* 1 × 10^5^, *E. necatrix* 1 × 10^5^, *E. acervulina* 1 × 10^5^, and *E. maxima* 1 × 10^5^) by oral gavage according to the method [21], and the NC group was given the same amount of normal saline in the same way. The temperature of the room was set at 33–35 °C during the first week, and then decreased by 2 °C every week to 24 °C. Feed intake on a replicate basis was evaluated every day, and body weight was measured on days 0, 14, and 42 at replicate level (10 broilers per replicate). Average daily gain (ADG), average daily feed intake (ADFI), and the feed/gain (F/G) ratio were calculated. The mortality of broilers was recorded daily.

### 2.4. Sample Collection

On days 21 and 42, one broiler from each replicate was randomly selected and killed by cervical dislocation. Blood samples were collected by cardiac puncture using vacuum tubes with coagulant and centrifuged at 3000 *g* for 10 min at 4 °C. The collected pure serum samples were stored in 1.5 mL Eppendorf tubes at −20 °C. The tissue samples of duodenum, jejunum and ileum were collected and fixed in 4% buffered formaldehyde. Cecal samples were harvested and stored at −20 °C for cecal lesion analysis.

### 2.5. Measurement of Anticoccidial Indicators

One sample from each replicate (6 broilers per treatment) were observed and collected on day 19 (5 days after challenge infection). Broiler excreta samples from each replicate were observed and collected on day 19 (5 days after challenge infection), and the bloody diarrhea score was scored on a scale of 0 to 4 according to More-house’s method [22], a score of 0 (no bloody feces contents), 1 (less than 25% feces contents), 2 (26–50% feces contents), 3 (51–75% feces contents), or 4 (over 75% feces contents) is recorded. The number of fecal oocysts was calculated according to McMaster’s method [23] and expressed as oocysts per gram of excreta (×10^5^/g of excreta). The cecal lesion score from six cecal samples per group was evaluated according to the method of Johnson and Reid [24], based on the macroscopic general appearance of petechial, thick, or shrunken intestinal walls, and bloody cecal contents, and a score of 0 (no lesions), 1 (mild lesions), 2 (moderate lesions), 3 (severe lesions), or 4 (extremely severe lesions or death due to coccidiosis) is recorded.

### 2.6. Analysis of D-Lactate, Diamine Oxidase (DAO), Immunity, and Antioxidant Status

The levels of *D*-lactic acid, DAO, malondialdehyde (MDA), interleukin 6 (IL-6), IL-10, tumor necrosis factor-α (TNF-α), and immunoglobulin A (IgA) in serum were determined using ELISA kits from Shanghai Enzyme-Linked Biotechnology Co., Ltd. (Shanghai, China) according to the manufacturer’s instructions (Appendix A). The activities of superoxide dismutase, catalase, glutathione peroxidase (GSH-Px), and total antioxidant capacity (T-AOC) in serum were determined using commercial kits from Suzhou Grise Biotechnology Co., Ltd. (Suzhou, China) according to the manufacturer’s instructions (Appendix A). There were serum samples from each replicate (6 broilers per treatment) and each sample was repeated 3 times.

### 2.7. Intestinal Morphology

The fixed intestinal segments from each replicate (6 broilers per treatment) in 4% formaldehyde were embedded in paraffin. Sections of each sample were stained with hematoxylin and eosin, and subsequently imaged with an Olympus microscope (Olympus, Tokyo, Japan) using the HMIAS-2000 image analysis system. Villus height was measured from the villus apex to the villus crypt junction, and crypt depth was measured from the base of the villus to the basolateral membrane. The villus height/crypt depth (V/C) value was calculated from these measurements [25].

### 2.8. Statistical Analysis

All statistical analyses were performed with the statistical software SPSS Statistics 20.0 (SPSS Statistics, Chicago, IL, USA). Normal distribution and homogeneity of variances were checked by the Shapiro–Wilk and Levene’s test, respectively. Growth performance, immune indices, antioxidant indices, *D*-lactate, DAO, and intestinal morphology data were analyzed using one-way ANOVA and anticoccidial indicators (oocysts per gram of excreta, cecal lesion score, and bloody diarrhea score) data were analyzed using the Kruskal–Wallis test. Duncan’s multiple comparison was used to compare the differences among the four groups. The replicate was considered as an experimental unit for statistical analysis of growth performance and oocysts per gram of excreta data. For the other data, the mean of 1 broiler per replicate served as an experimental unit for statistical analysis. A *p*-value of 0.05 was considered statistically significant.

## 3. Results

### 3.1. Effects of CGA on the Growth Performance and Mortality of Coccidia-Infected Broilers

No significant differences were observed in the ADG, ADFI, and F/G ratio among the groups during days 0 to 14 (Table 2). During days 14 to 42, broilers in the NC and CGA1 group had a higher (*p* < 0.001) ADG and a lower (*p* = 0.010) F/G ratio than those in the IC group. In addition, mortality in the IC group was higher (*p* = 0.030) than that in the NC and CGA1 groups, but no significant differences were observed among the NC, CGA0.5, and CGA1 groups.

### 3.2. Effects of CGA on the Anticoccidial Indicators of Coccidia-Infected Broilers

The oocysts per gram of excreta in the IC group was higher (*p* < 0.001) than that in the NC and CGA1 groups, but no significant differences were observed among the NC, CGA0.5, and CGA1 groups (Table 3). The cecal lesion score in the NC group was lower (*p* = 0.001) than that in the other groups, but no significant differences were observed among the IC, CGA0.5 and CGA1 groups. The bloody diarrhea score in the IC group was higher (*p* = 0.001) compared with the NC and CGA1 groups.

### 3.3. Effects of CGA on Immune Indices in the Serum of Coccidia-Infected Broilers

On day 21, the IL-6 level in the IC group was higher (*p* = 0.002) than that in the other groups, but no significant differences were observed among the NC, CGA0.5, and CGA1 groups (Table 4). Broilers in the IC group had lower IL-10 (*p* = 0.027) and IgA (*p* = 0.042) levels than those in the NC group, but no significant differences were observed among the NC, CGA0.5, and CGA1 groups. On day 42, broilers in the IC group had a higher (*p* = 0.048) IL-6 level and a lower (*p* = 0.017) IL-10 level than those in the NC group, but no significant differences were observed among the NC, CGA0.5, and CGA1 groups.

### 3.4. Effects of CGA on Antioxidant Indices in the Serum of Coccidia-Infected Broilers

On day 21, the T-AOC activity in the IC group was lower (*p* < 0.001) than that in the other groups, but no significant differences were observed among the NC, CGA0.5, and CGA1 groups (Table 5). The MDA level in the IC group was higher (*p* < 0.001) than that in the other groups. On day 42, compared with the IC and CGA0.5 groups, the GSH-Px activity was higher (*p* < 0.001) in the NC group, but no significant differences were observed between the NC and CGA1 groups. The MDA level in the IC group was higher (*p* < 0.001) than that in the other groups, but no significant differences were observed among the NC, CGA0.5, and CGA1 groups.

### 3.5. Effects of CGA on the Levels of D-Lactate and DAO in the Serum of Coccidia-Infected Broilers

On day 21, the *D*-lactic acid level in the IC group was higher (*p* < 0.001) than that in the other groups, but no significant differences were observed among the NC, CGA0.5, and CGA1 groups (Table 6). The DAO level in the IC group was higher (*p* < 0.001) than that in the other groups. On day 42, the *D*-lactic acid level in the IC group was higher (*p* = 0.003) than that in the other groups, but no significant differences were observed among the NC, CGA0.5, and CGA1 groups. Compared with the IC group, the DAO level was higher (*p* < 0.001) in the NC and CGA1 groups.

### 3.6. Effects of CGA on the Intestinal Morphology of Coccidia-Infected Broilers

On day 21, compared with the IC group, villus height in the duodenum was higher (*p* < 0.001) in the NC and CGA1 groups, but no significant differences were observed between the CGA0.5 and CGA1 groups (Table 7 and Figure 1A). Compared with the IC group, villus height in the jejunum was higher (*p* = 0.017) in the NC and CGA1 groups. On day 42, compared with the IC group, villus height in the duodenum and jejunum was higher (*p* < 0.001) in the other groups (Figure 1B). Compared with the IC group, villus height in the ileum was higher (*p* < 0.001) in the NC and CGA1 groups. The V/C value of the duodenum in the IC and CGA0.5 groups was lower (*p* = 0.017) than that in the NC group, but no significant differences were observed between the NC and CGA1 groups.

## 4. Discussion

Growth performance, intestinal lesions, oocysts per gram of excreta, and bloody diarrhea are the main parameters to evaluate the severity of coccidia infections in broilers [26]. In the present study, coccidia infection increased oocysts per gram of excreta, bloody diarrhea score, and cecal lesion score, which was consistent with the report by Mansoori et al. [27], who found that coccidia infection increased oocyst shedding and lesion scores in broilers on 5 days after infection. In previous studies, excreta were collected for 9 consecutive days after coccidia infection in broilers. The number of oocysts per gram of excreta in broilers increased with the number of infected days, and the study found that *Emblica officinalis* derived tannins had a better protective effect in coccidia-infected broilers [28]. In this study, dietary supplementation with 1 g/kg CGA decreased the oocysts per gram of excreta and bloody diarrhea score in coccidia-infected broilers on the 5 days after infection, indicating that CGA inhibited the degree of coccidia infection in broilers. However, the effect of CGA on coccidia could not be determined by fecal oocytes at one-time point in this study. This is the limitation of this study, which requires further study and explanation. Moreover, coccidia infection decreased the growth performance of broilers in our study, which was in line with the results of El-Shazly et al. [29], who reported that coccidia infection decreased the ADG and increased the F/G ratio and mortality in broilers. Zhang et al. [17] reported that dietary supplementation with CGA alleviated depressed growth performance of broilers with necrotic enteritis, which was similar to our results where dietary supplementation with 1 g/kg CGA significantly increased the ADG and decreased the F/G ratio and mortality in coccidia-infected broilers. These results implied that CGA improved growth performance by decreasing oocysts per gram of excreta, bloody diarrhea score, and cecal lesion score in coccidia-infected broilers.

Coccidiosis infection can induce inflammatory responses, resulting in increased levels of pro-inflammatory factors such as IL-6 and TNF-α in broilers [30,31]. In the present study, coccidiosis infection increased the IL-6 level and decreased the IL-10 and IgA levels, indicating that coccidiosis infection could induce inflammatory responses. However, we noted that dietary supplementation with CGA alleviated systemic inflammatory responses in coccidia-infected broilers on day 21, as dietary supplementation with CGA decreased IL-6 level and no significant differences were observed in IL-6, IL-10, and IgA levels among the NC, CGA0.5, and CGA1 groups on day 21. Similar results were found in a study by Peng et al. [32], who demonstrated that CGA-enriched extract decreased IL-6 and TNF-α levels in 67-week-old hens.

Oxidative stress may increase the level of MDA and decrease antioxidant enzyme activities in broilers [33]. In the present study, coccidiosis infection increased MDA level, and decreased T-AOC and GSH-Px activities, which was similar to the results of Fortuoso et al. [34]. These results indicated that coccidial infection induced oxidative stress. Moreover, dietary supplementation with CGA increased T-AOC activity on day 21 and decreased MDA level on days 21 and 42 in the serum of broilers, indicating that it effectively alleviated the oxidative stress caused by coccidial infection. Chen et al. found [35] that dietary supplementation with CGA could increase antioxidant capacity and alleviate oxidative stress damage caused by high temperature in young hens.

Villus height, crypt depth, and V/C value can reflect the integrity, development status, and nutrient absorption capacity of the intestine in animals [36]. In the current study, coccidia infection decreased villus height in broilers, which was consistent with the results reported in previous studies [37]. In addition, we found that dietary supplementation with 1 g/kg CGA increased the villus height caused by coccidia infection in this study, which was consistent with the report by Zhang et al. [17], who found that dietary supplementation with CGA could improve villus height and maintain the intestinal integrity of broilers with necrotic enteritis. The levels of DAO and *D*-lactic acid in serum can be used as markers to monitor intestinal permeability and barrier damage [38]. Studies suggest that pathogen and coccidia infection can affect intestinal barrier function, which will increase the serum DAO and *D*-lactic acid levels in broilers [39,40,41]. In our study, coccidia infection increased the levels of DAO and *D*-lactic acid in serum, while dietary supplementation with 0.5 and 1 g/kg CGA decreased the levels of DAO on days 21 and 42 and *D*-lactic acid on day 21, and dietary supplementation with 1 g/kg CGA decreased the levels of DAO on day 42. These results indicated that dietary supplementation with CGA could promote growth performance and alleviate intestinal barrier function damage by improving intestinal permeability and intestinal morphology in coccidia-infected broilers. 

## 5. Conclusions

Dietary supplementation with CGA decreased oocysts per gram of excreta, cecal lesion score, and bloody diarrhea score, and improved growth performance, immunity, antioxidant status, and intestinal barrier function in coccidia-infected broilers. Considering all these indices, CGA could be used as a potential substance to alleviate the effect of coccidia in poultry production, and dietary supplementation with CGA at 1 g/kg alleviated the effect of coccidia in broilers.

## Figures and Tables

**Figure 1 animals-12-00963-f001:**
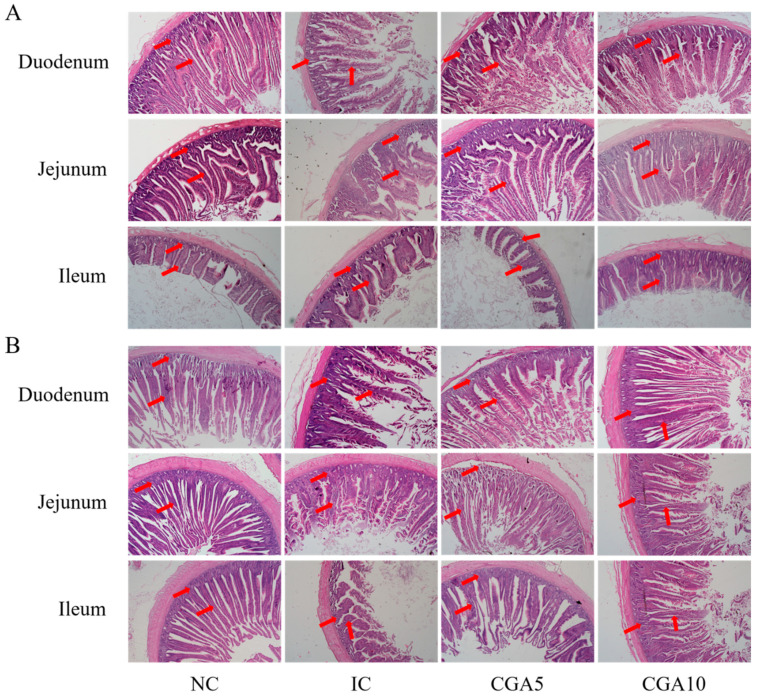
Effects of chlorogenic acid (CGA) on the intestinal morphology of broilers (*n* = 6). (**A**) Day 21, (**B**) day 42. NC: Control diet; IC: Control diet + *Eimeria* infection; CGA0.5: Control diet + 0.5 g/kg CGA + *Eimeria* infection; CGA1: Control diet + 1 g/kg CGA + *Eimeria* infection. Images in the transverse section (Hematoxylin and Eosin stained) were captured at 100× magnification.

**Table 1 animals-12-00963-t001:** Ingredients and nutrient levels in the basal diet.

Item	Contents	
	1–21 Days of Age	22–42 Days of Age
Ingredients (%, as fed)		
Corn	60.00	62.00
Soybean meal	34.30	30.50
Soybean oil	2.00	4.00
Limestone	1.45	1.40
CaHPO_4_	1.33	1.28
*DL*-Methionine	0.25	0.15
NaCl	0.35	0.35
Premix ^1^	0.20	0.20
Multi-vitamin ^2^	0.02	0.02
Cholione chloride	0.10	0.10
Total	100.00	100.00
Total Nutrient levels ^3^ (%, as fed)		
Crude protein	20.65	18.98
Calcium	1.00	0.90
Available phosphorus	0.45	0.40
Lysine	1.09	0.99
Methionine	0.56	0.44
Metabolizable energy (MJ/kg)	12.54	12.96

^1^ Provided per kilogram of diet: Fe (as ferrous sulfate) 80 mg; Cu (as copper sulfate) 10 mg; Zn (as zinc sulfate) 75 mg; Mn (as manganese sulfate) 80 mg; Se (as sodium selenite) 0.30 mg; I (as potassium iodide) 0.40 mg. ^2^ Provided per kilogram of diet: Vitamin A (trans-retinyl acetate) 8000 IU; Vitamin D_3_ (cholecalciferol) 3000 IU; Vitamin E (all-rac-α-tocopherol acetate) 20 IU; Vitamin K_3_ (menadione) 2.0 mg; Vitamin B_1_ (thiamin) 4.2 mg; Vitamin B_2_ (riboflavin) 4.0 mg; Vitamin B_6_ (pyridoxine HCl) 4.5 mg; Vitamin B_12_ (cobalamin) 0.02 mg; nicotinic acid 10 mg; calcium pantothenate 11 mg; folic acid 1.0 mg; biotin 0.15 mg. ^3^ The nutrient levels were calculated values.

**Table 2 animals-12-00963-t002:** Effects of chlorogenic acid (CGA) on the growth performance of broilers (*n* = 6).

Item ^1^	NC	IC	CGA0.5	CGA1	SEM ^2^	*p*-Value
Days 0 to 14						
Average daily gain (g)	20.58	21.13	20.96	22.07	0.25	0.183
Average daily feed intake (g)	33.88	35.55	34.01	35.59	0.46	0.378
Feed/gain ratio	1.65	1.69	1.63	1.61	0.02	0.683
Days 14 to 42						
Average daily gain (g)	58.41 ^a^	47.33 ^c^	50.04 ^bc^	53.28 ^b^	0.94	<0.001
Average daily feed intake (g)	98.82	94.16	93.80	92.38	1.59	0.522
Feed/gain ratio	1.70 ^b^	2.00 ^a^	1.88 ^ab^	1.74 ^b^	0.04	0.010
Mortality (%)	0 ^b^	11.69 ^a^	7.79 ^ab^	2.60 ^b^	0.02	0.030

^a,b,c^ Mean value within row with no common superscript differ significantly (*p* < 0.05). ^1^ NC: Control diet; IC: Control diet + *Eimeria* infection; CGA0.5: Control diet + 0.5 g/kg CGA + *Eimeria* infection; CGA1: Control diet + 1 g/kg CGA + *Eimeria* infection. ^2^ Standard error means.

**Table 3 animals-12-00963-t003:** Effects of chlorogenic acid (CGA) on the anticoccidial indicators of broilers (*n* = 6).

Item ^1^	NC	IC	CGA0.5	CGA1	SEM ^2^	*p*-Value
5 days after infection						
Oocysts per gram of excreta ^3^ (× 10^5^/g of excreta)	0.00 ^c^	13.55 ^a^	4.66 ^ab^	1.52 ^b^	4.05	<0.001
Cecal lesion score ^4^	0.00 ^b^	2.33 ^a^	1.83 ^a^	1.50 ^a^	3.95	0.001
Bloody diarrhea score ^5^	0.00 ^c^	1.50 ^a^	0.92 ^ab^	0.67 ^b^	3.96	0.001

^a,b,c^ Mean value within row with no common superscript differ significantly (*p* < 0.05). ^1^ NC: Control diet; IC: Control diet + *Eimeria* infection; CGA0.5: Control diet + 0.5 g/kg CGA + *Eimeria* infection; CGA1: Control diet + 1 g/kg CGA + *Eimeria* infection. ^2^ Standard error means. ^3^ The oocytes of excreta are observed and counted by microscope. ^4^ A score of 0 (no bloody feces contents), 1 (less than 25% feces contents), 2 (26–50% feces contents), 3 (51–75% feces contents), or 4 (over 75% feces contents). ^5^ A score of 0 (no lesions), 1 (mild lesions), 2 (moderate lesions), 3 (severe lesions), or 4 (extremely severe lesions or death due to coccidiosis) is recorded.

**Table 4 animals-12-00963-t004:** Effects of chlorogenic acid (CGA) on immune indices in the serum of broilers (*n* = 6).

Item ^1^	NC	IC	CGA0.5	CGA1	SEM ^2^	*p*-Value
Day 21						
Interleukin 6 (pg/mL)	28.64 ^b^	40.20 ^a^	33.43 ^b^	32.08 ^b^	1.32	0.002
Interleukin 10 (pg/mL)	106.46 ^a^	78.74 ^b^	85.16 ^ab^	87.79 ^ab^	3.48	0.027
Tumor necrosis factor-α (pg/mL)	49.10	63.86	56.35	55.60	2.08	0.154
Immunoglobulin A (μg/mL)	292.24 ^a^	240.91 ^b^	253.07 ^ab^	250.04 ^ab^	8.09	0.042
Day 42						
Interleukin 6 (pg/mL)	25.58 ^b^	33.81 ^a^	27.47 ^ab^	26.90 ^ab^	1.27	0.048
Interleukin 10 (pg/mL)	107.41 ^a^	62.23 ^b^	95.45 ^ab^	91.26 ^ab^	4.66	0.017
Tumor necrosis factor-α (pg/mL)	50.17	64.12	58.32	55.08	1.95	0.123
Immunoglobulin A (μg/mL)	255.24	232.14	247.44	243.11	5.38	0.598

^a,b^ Mean value within row with no common superscript differ significantly (*p* < 0.05). ^1^ NC: Control diet; IC: Control diet + *Eimeria* infection; CGA0.5: Control diet + 0.5 g/kg CGA + *Eimeria* infection; CGA1: Control diet + 1 g/kg CGA + *Eimeria* infection. ^2^ Standard error means.

**Table 5 animals-12-00963-t005:** Effects of chlorogenic acid (CGA) on antioxidant indices in the serum of broilers (*n* = 6).

Item ^1^	NC	IC	CGA0.5	CGA1	SEM ^2^	*p*-Value
Day 21						
Total antioxidant capacity (U/mL)	0.77 ^a^	0.57 ^b^	0.72 ^a^	0.74 ^a^	0.02	<0.001
Catalase (U/mL)	42.44	38.96	38.77	42.01	0.98	0.534
Glutathione peroxidase (U/mL)	913.94	807.55	892.10	916.27	15.69	0.083
Superoxide dismutase (U/mL)	175.12	166.10	168.93	165.15	2.00	0.286
Malondialdehyde (nmol/mL)	0.65 ^c^	1.15 ^a^	0.91 ^b^	0.85 ^bc^	0.05	<0.001
Day 42						
Total antioxidant capacity (U/mL)	0.83	0.70	0.72	0.77	0.02	0.313
Catalase (U/mL)	43.21	41.36	41.18	41.42	0.85	0.858
Glutathione peroxidase (U/mL)	986.97 ^a^	910.48 ^b^	924.51 ^b^	946.34 ^ab^	8.56	<0.001
Superoxide dismutase (U/mL)	173.82	166.82	168.47	170.80	1.34	0.354
Malondialdehyde (nmol/mL)	0.77 ^b^	1.01 ^a^	0.80 ^b^	0.74 ^b^	0.02	<0.001

^a,b,c^ Mean value within row with no common superscript differ significantly (*p* < 0.05). ^1^ NC: Control diet; IC: Control diet + *Eimeria* infection; CGA0.5: Control diet + 0.5 g/kg CGA + *Eimeria* infection; CGA1: Control diet + 1 g/kg CGA + *Eimeria* infection. ^2^ Standard error means.

**Table 6 animals-12-00963-t006:** Effects of chlorogenic acid (CGA) on serum *D*-lactate and DAO levels of broilers (*n* = 6).

Item ^1^	NC	IC	CGA0.5	CGA1	SEM ^2^	*p*-Value
Day 21						
*D*-lactic acid (μmol/L)	53.76 ^b^	77.82 ^a^	64.51 ^b^	65.80 ^b^	2.20	<0.001
Diamine oxidase (ng/mL)	19.55 ^c^	30.68 ^a^	24.79 ^b^	23.64 ^b^	0.82	<0.001
Day 42						
*D*-lactic acid (μmol/L)	49.87 ^b^	76.88 ^a^	57.81 ^b^	56.80 ^b^	2.60	0.003
Diamine oxidase (ng/mL)	16.89 ^c^	24.51 ^a^	24.39 ^a^	20.83 ^b^	0.72	<0.001

^a,b,c^ Mean value within row with no common superscript differ significantly (*p* < 0.05). ^1^ NC: Control diet; IC: Control diet + *Eimeria* infection; CGA0.5: Control diet + 0.5 g/kg CGA + *Eimeria* infection; CGA1: Control diet + 1 g/kg CGA + *Eimeria* infection. ^2^ Standard error means.

**Table 7 animals-12-00963-t007:** Effects of chlorogenic acid (CGA) on the intestinal morphology of broilers (*n* = 6).

Item ^1^	NC	IC	CGA0.5	CGA1	SEM ^2^	*p*-Value
Day 21						
Duodenum						
Villus height (μm)	783.62 ^a^	628.23 ^c^	650.17 ^bc^	682.11 ^b^	19.05	<0.001
Crypt depth (μm)	113.02	104.62	109.06	105.42	1.96	0.462
Villus height/crypt depth	6.94	6.01	5.96	6.52	0.18	0.098
Jejunum						
Villus height (μm)	316.64 ^a^	263.57 ^b^	272.61 ^b^	289.55 ^a^	7.27	0.017
Crypt depth (μm)	65.56	75.96	72.76	66.93	1.89	0.377
Villus height/crypt depth	4.83	3.47	3.77	4.34	0.22	0.061
Ileum						
Villus height (μm)	179.27	139.42	148.94	157.78	6.76	0.189
Crypt depth (μm)	50.13	48.62	51.44	49.70	1.79	0.969
Villus height/crypt depth	3.63	2.95	2.92	3.17	0.18	0.504
Day 42						
Duodenum						
Villus height (μm)	940.03 ^a^	765.20 ^c^	811.48 ^b^	920.73 ^a^	22.46	<0.001
Crypt depth (μm)	151.07	154.20	152.95	153.05	2.28	0.972
Villus height/crypt depth	6.22 ^a^	4.97 ^b^	5.32 ^b^	5.61 ^ab^	0.17	0.017
Jejunum						
Villus height (μm)	617.64 ^a^	558.13 ^c^	576.79 ^b^	585.01 ^b^	6.78	<0.001
Crypt depth (μm)	102.13	106.16	107.15	108.36	1.89	0.736
Villus height/crypt depth	6.05	5.27	5.39	5.44	0.13	0.068
Ileum						
Villus height (μm)	391.71 ^a^	336.55 ^c^	346.08 ^c^	372.02 ^b^	7.03	<0.001
Crypt depth (μm)	53.96	49.77	50.51	53.72	1.43	0.699
Villus height/crypt depth	7.30	6.91	6.87	6.95	0.20	0.902

^a,b,c^ Mean value within row with no common superscript differ significantly (*p* < 0.05). ^1^ NC: Control diet; IC: Control diet + *Eimeria* infection; CGA0.5: Control diet + 0.5 g/kg CGA + *Eimeria* infection; CGA1: Control diet + 1 g/kg CGA + *Eimeria* infection. ^2^ Standard error means.

## Data Availability

All data sets collected and analyzed during the current study are available from the corresponding author on fair request.

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
