# Peer review of "Effects of Chlorogenic Acid on Performance, Anticoccidial Indicators, Immunity, Antioxidant Status, and Intestinal Barrier Function in Coccidia-Infected Broilers"

_animals, 2022, doi:10.3390/ani12080963_

Round 1
Reviewer 1 Report
Dear authors,
Reviewer comments: The focus of the article entitled “Effects of Chlorogenic Acid on Performance, Anticoccidial Indicators, Immunity, Antioxidant Status and Intestinal Barrier Function in Coccidia-Infected Broilers” is to know more about the performance and intestinal health on avian coccidiosis by dietary supplementation with chlorogenic acid. It is a very attractive study with interesting results and conclusions.
General comments
Since each Eimeria species has different pathogenicity, and tropism for intestinal tract (duodenum, jejunum, ileum and cecum in the case of E. necatrix), it would be interesting to add information about differences in OPG of the four species excreted in faeces after the infection related to the intestinal morphology results. Did the authors sporulate and identificate the oocysts?
Abstract
Line 23. I suggest using lower case to write “Infected control”, in accordance with “non-infected (same for line 82, materials and methods section).
Introduction
Line 46. Write E. tenella, E. necatrix, E. acervulina, E. maxima (same for line 71, materials and methods section)
References
Ref. no 27 is strikethrough.
Figures and tables
Figure 1. The images are quite small, and it is difficult to appreciate some details. Add info in the legend regarding the stain and the magnification.
Tables 2 to 7. Change 2SEM for SEM2, or unify with the item in tables.
Reviewer 2 Report
Comments to Authors
General comments
The manuscript titled “Effects of Chlorogenic Acid on Performance, Anticoccidial Indicators, Immunity, Antioxidant Status and Intestinal Barrier Function in Coccidia-Infected Broilers” is a clinical trial that investigated the effect of chlorogenic acid as a natural plant dietary supplementation to alleviate the damage associated with coccidia infection in broilers. The research design is appropriate, and the results were clearly presented. The manuscript is well written and easy to follow. However, there are issues that need to be fixed before the paper gets published. For example,
- Authors have to be careful in their conclusions when stating that chlorogenic acid (CGA) has an anticoccidial effect since the study didn’t investigate the toxic effect of CGA on Eimeria Also, the study investigated the effect of CGA on fecal oocytes output at a one-time point (5 days post-infection) which is not enough to conclude that CGA has an anticoccidial effect. Therefore, the authors should add a section about the limitations of this study.
- In many cases within the paper, the conclusions are not supported by the results. For example, the authors mentioned that “we found that dietary supplementation with CGA increased the villus height and V/C value caused by coccidia infection in this study” L251-252. This is incorrect since the study results showed that
- On day 21, villus height was not significantly different between IC and CGA5. Also, the V/C value was not significantly different among all groups.
-Day 42, V/C value was not significantly different among all treatment groups.
Please correct your conclusions.
- Detailed information is missing from the materials and methods.
Please see attached pdf with more specific comments that need to be addressed.
Thank you.

Round 2
Reviewer 1 Report
Dear authors, in my opinion the manuscript has improved after the suggested changes and can be accepted for publication in present form.
Reviewer 2 Report
Thanks authors for addressing my comments. You did a great job which improved the quality of the manuscript. Please consider English proofreading of the manuscript before being published.